# Students' negative emotions and their rational and irrational behaviors during COVID-19 outbreak

**Mahdi Rezapour**[1]*, **Arash Dehzangi**[2], **Farzaneh Saadati**[3]

**1** Independent researcher, Iran, **2** Department of Electrical and Computer Engineering, Northwestern University, Evanston, IL, United States of America, **3** Center for Advanced Research in Education (CIAE), Universidad de Chile, Santiago, Chile

* Rezapour2088@yahoo.com

## Abstract

The pandemic has posed an intense threat to the mental health of younger adults. Despite significant efforts in studying various aspects of COVID-19, there is a dearth of evidence on how negative emotions are associated with behaviors. A comparison across associated factors to different negative emotions by means of a unified model is especially missing from the literature. This study was conducted by using the results of a survey conducted across 2,534 students enrolled in 7 states in the US. Various feelings such as sad, irritable, stress and guilt were analyzed in a unified model by means of seemingly unrelated regression (SUR). Questions were asked related to rationally limiting the spread of virus, and questions related to behaviors that seem to be extreme. Irrational behaviors, such as limiting exercise at home due to COVID-19 could be due to experiencing negative emotions, which distort the meaning of events for the students. That behavior, for instance, was found to be positively associated with various negative feelings. In addition, the results highlighted significant differences across emotions in terms of demographic characteristics such as gender and age, and various precautionary actions that students take, such as limiting outdoor activities or limiting social gathering. For instance, it was highlighted while avoiding a large group of people, in spite of its importance in curbing the spread of virus, is negatively associated with various negative emotions, taking a possible nonconsequential precaution of completely limiting all people outside the immediate family result in a higher level of negative emotions. Also, it was found that having a higher body mass index (BMI), self-rated worse health conditions, and limiting outdoor activities have detrimental effects on the mental health of students.

## Introduction

Countries around the world have responded to curb the spread of the virus COVID-19 by imposing strict measures such as lockdown or social distancing. This situation has led to changes in education systems so that schools and colleges have been subjected to different adjustments and closing in more than 180 countries [1].

**Data Availability Statement:** The data could be found in paper titled Psychological impacts from COVID-19 among university students: Risk factors across seven states in the United States at the link

of https://journals.plos.org/plosone/article?id=10.1371/journal.pone.0245327.

**Funding:** The author(s) received no specific funding for this work.

**Competing interests:** The authors have declared that no competing interests exist.

Although strict measures such as social distancing is required to contain the spread of the disease, it impacts the individuals in terms of their habits, lifestyles, and how they see things during this pandemic. COVID-19 resulted in a rapidly increased number of people experiencing psychological problems such as depression, stress and anxiety [2], especially among college students [3]. Since starting the pandemic, several studies and different aspects of associated negative emotions related to COVID-19 has been reported in the literature. So far, extensive works have been employed for various aspects of COVID-19, from immunity [5], to its variant [6]. Some analysis reveals that the impact of the pandemic on student learning at different level of education was significant, leaving them behind in educational goals and learning. For instance, the negative impact of social distancing in terms of anxiety [4], stress and anxiety due to the pandemic [5], and emotional intelligence across front-line nurses [6] were evaluated and reported by different groups.

Besides the common negative emotions that the pandemic is expected to impose on the well-being of people, it is also expected to impose some negative impacts that are not related to the containment of virus. There is a distorted perception regarding events or the consequences of previous negative feelings, which would turn into a behavior. For instance, it has been discussed that negative emotions could distort the meaning of events, so people interpret their experiences about an event in a negative or self-defeating way [7].

Studying emotions is especially important as they are expected to impact the general health of people. Previous studies emphasized on importance of positive emotions on promoting health [8], and how the absence of negative emotions can result in better health and longevity [9].

In general, the role of emotions in students' learning and academic achievement has received increasing attention in recent years [10].

However, more emphasis has been placed on anxiety and its impact on students' achievements in exams. While less attention is paid to the effects of a wider range of negative emotions such as sadness, grief, boredom and anger in students' lives and their behaviors. In the present study, we are studying a wide range of emotions such as sadness, fear, irritability, stress and guilt, where associations between these emotions and various precautionary behaviors, rational and irrational, are considered. This study is also going to conduct an evaluation of the associations between various emotions that students might experience during the COVID-19 pandemic and other student characteristics and behavioral actions. Emotions of sadness, fear, irritability, stress and guilt were incorporated in a unified model by means of seemingly unrelated regression (SUR). In this survey, questions regarding to rational and irrational behaviors were incorporated to study the behavioral reactions of students during the pandemic.

## Literature review

In this study feelings of sadness, fear, guilt, irritability, stress were considered as the response, which the following paragraphs will briefly outline.

Sadness was discussed as universal and an inevitable aspect of human beings, which might be a response to the death of a loved one [11]. Sadness in literature has been mainly looked at from a depression perspective and anxiety. For instance, a study found an association between major life experience, and life stress and depression [12]. Researchers who worked with students also pay more attention about the relationship between anxiety and final school tests [10].

Feeling of fear occurs when one is in danger, and it controls defensive response to threats [13]. In another study, feeling fear has been found to be related to a sense of insecurity in unfamiliar circumstances [14].

On the other hand, irritability could be defined as conceptualized anger associated with other negative emotions, which could occur with minimum provocation [15]. Irritability has been also discussed as a non-specific manifestation of a psychiatric disturbance such as reaction to pain or a mood state, being independent of other emotions such as anxiety [16]. Irritability has been named as the precursors for anger and aggression [17]. Some of the main causes of irritably has been discussed as feeling pressured, waiting and being uncertain about the future or an answer to the problem at hand [18].

Feeling stressed has been defined as a body's response to perceived threats, which might be in reaction to environmental exposures [19]. The physiological changes, resulted from stress, might be protective, which prepare the individuals in response to danger or threat [20].

On the other hand, the importance of guilt and shame for the socialization process have been evaluated quite closely (14); it was discussed that the concept of self-concept is the basis for feeling of guilt. In another study, guilt as a negative emotion has been defined as a self-evaluative process, which arises from observed behavior in conflict with ones' understanding of social norms [21].

On the other hand, emotions have been defined as psychological states which are accompanied by behaviors, which in turn originated from evaluation or appraisal regarding a change in environment or a current circumstance [9]. At the same time, intrapsychic and mind processes determine the choices we make and how we act as a behavioral perspective [22]. For instance, a past study found a relationship between emotions and the performance of work [23]. Also, studies investigated irrational and aberrant behaviors during crisis. For instance, irrational stockpiling was noted during COVID-19 [24].

The first precaution was taken against spreading of the COVID-19 virus was social distancing and isolation, which caused fear and stress (due to unknown illness), depression, loss of normal life and human communication. It was explained that these negative emotions are associated with decreased levels of physical activity during the pandemic [25]. Considering students' life which can always challenge their mental well-being, studies showed that the pandemic has indeed disrupted students' lives and created sense of uncertainty and stress with unfavorable effects on their learning and mental well-being [25].

In this light, it is important to improve our understanding about the negative emotions and their relationships with students' behavior in order to get prepared for helping students in the post-pandemic era.

As a result, based on the current literature, we cannot certainly conclude that how the associated factors of various negative emotions vary. To gain an insight regarding this uncertainly, we model the emotions in a unified modeling framework so comparison could be made for the magnitudes and signs of different variables.

## Methodology and data analysis

The surveys were distributed across 14,174 students enrolled in seven U.S. universities, including Arizona State University in Tempe, Clemson University in Clemson, North Carolina State University in Raleigh, Oregon State University, Pennsylvania State University, University of Montana, and University of Utah. Out of those, 2,534 responses were obtained and used [26]. From those responses 61% were female compared with 39% being male, also 77% were younger students (i.e., 18–24 years old), and the rest were older. Various questions were used in the survey which are mostly self-explanatory and presented in Table 1. Body mass index (BMI) was calculated from the self-reported height and weight on a 4-point scale, from excellent to poor (obese).

For the quantitative assessment, nine survey items were chosen based on information gathered from all the data, and all the concepts of interest were measured from these survey items.

**Table 1. Descriptive statistics of important predictors considered in the analysis.**

| Coef. | Mean | Var | Min | Max | Type |
|---|---|---|---|---|---|
| *Response* | | | | | |
| How sad do you feel when you think about COVID-19? | 61.12 | 725.575 | 0 | 100 | Likert, Continuous |
| How afraid do you feel when you think about COVID-19? | 50.55 | 742.89 | 0 | 100 | Likert, Continuous |
| How irritable do you feel when you think about COVID-19? | 59.43 | 791.26 | 0 | 100 | Likert, Continuous |
| How stressed do you feel when you think about COVID-19? | 64.03 | 693.92 | 0 | 100 | Likert, Continuous |
| How guilty do you feel when you think about COVID-19? | 24.36 | 657.680 | 0 | 100 | Likert, Continuous |
| *Other explanatory variables* | | | | | |
| Gender, male*, versus female | 0.61 | 0.233 | 0 | 1 | Binary |
| Respondents' highest level of education, graduate versus under grad* | 0.196 | 0.158 | 0 | 1 | Binary |
| Age, 18–24 years old, versus others* | 0.77 | 0.178 | 0 | 1 | Binary |
| Health in general | 3.32 | 1.03 | 1 | 5 | Likert |
| BMI group | 2.38 | 0.490 | 1 | 4 | Likert |
| Precautionary actions: avoid travel | 4.52 | 0.616 | 1 | 5 | Likert |
| Precautionary actions: limit outdoor activities | 3.15 | 1.727 | 1 | 5 | Likert |
| Precautionary actions: limit exercising at home | 2.32 | 1.538 | 1 | 5 | Likert |
| Precautionary actions: Avoiding large group of people | 4.16 | 0.793 | 1 | 5 | Likert |
| Precautionary actions: avoid people outside of your immediate family | 4.72 | 0.416 | 1 | 5 | Likert |
| Spend a lot of time thinking about COVID-19 | 4.34 | 3.051 | 1 | 7 | Likert |
| Income relative to others in the US | 3.31 | 1.224 | 1 | 5 | Likert |
| Any infected individual among friends/family, yes versus no* | 0.25 | 0.176 | 0 | 1 | Binary |
| *Smoothed variables* | | | | | |
| Number of hours in front of the screen | 7.74 | 7.21 | 0 | 12 | Discrete |
| Limiting exercise at the gym | 4.65 | 0.938 | 0 | 5 | Discrete |

* Reference category.

The negative emotion states were four items of those, where each item was dedicated to each of the negative emotions. These negative emotions for each survey item were recognized during the development of the positive and negative affect schedule (PANAS) [27]. We had different emotions that the respondents experienced due to COVID-19 where all of those emotions were considered. In the data analysis, these negative emotions including being sad, afraid, irritable, stressed and guilty, with similar scale were used.

Students were asked about their "health in general" on a 5-point response scale from poor to excellent [26]. The question was based on self-rated general health [28]. Respondents' highest level of education was discovered by asking the students whether they are graduate or undergraduate students. In another question, the respondents were asked to rate the question of "I spend a lot of time thinking about COVID-19" from 1 (strongly disagree)-7 (strongly disagree). Family income, relative to others in the US, was measured from 1 (well below average)-5 (well above average). Students were also asked if they knew anyone infected by COVID-19 in their local community. To indicate various negative emotional feelings, the respondents could slide a vertical bar to the left or right to indicate their response, and it did not have to be a fixed value.

It should be noted compared with the previous study which used the similar dataset [26], we employed a unified technique to account for all types of negative emotions, instead of focusing on a single one. Also, the previous study focused on the structure of the variables by means of factor analysis, while we focused mainly on the impact of various variables in a unified model. The vector generalized linear model (VGLM) function in R was used for conducting the statistical analysis [29].

The criteria for selection of various responses for different models are due to specific question related to different negative emotions. In other words, all negative emotions highlighted in the survey were incorporated as our response for the statistical modeling. We used the smoothed parameters in the context of SUR as specific statistical method for analyzing the data.

Having SUR could be elaborated by stacking the equations for creating matrices, in terms of responses and explanatory variables, and then implementation of the smooth parameters on the prepared matrices, which the next few paragraphs outline. To prepare SUR, various equations are tied/stacked together as:

$$(y_1 | \ldots | y_M) = (X_1 \beta_1 | \ldots | X_M \beta_M) + (\varepsilon_1 | \ldots | \varepsilon_M) \tag{1}$$

where $M$ is the number of considered equations or responses, $\varepsilon_M$ is the error terms of equation $M$, and $y_M$, $X_M$ and $\beta_M$ are response, explanatory variables and parameter to be estimated, respectively. Correlations across errors terms in different equations are, $Cov(\varepsilon_n, \varepsilon_m) = \sigma_{nm}$.

The process of data preparation is based on pre-multiplication of both sides of equations by Cholesky decomposition of weight ($W$), where $W = U^T U$, so the ordinary least square (OLS) will be converted into generalized least square (GLS) by removing correlation between the error terms.

During the initialization process, the working response or $z$ is written based on $\eta$, weight and $u$ as: $z_i = \eta_i + W_i^{-1} u_i$. Then, consistently $z$ will be multiplied, and later be back transformed by $u$, and regressed against $x^*$, or the transformed explanatory variables. So, if we want to estimate $\beta$s we multiply $z$ by $u$, and regress against $x^*$ for estimation of $\beta$. While, checking the convergence, we back transform $z$ and check it against the response.

For creating the smooth parameters, values closer to the mean of the parameter are used and repetitive and further values from the mean ignored. That is based on the size of window which highlight how smooth or wiggly the smooth should be.

For the smoothing of the parameters, the variables are regularly compressed based on the smooth parameters and returned to the normal values to solve the backfitting algorithm. One of the advantages of the smoothing is that it does not suffer from the curse of dimensionality [30], as smooths are treated univariately. So, while smoothing a covariate all other covariates are kept as fixed.

$y$ in $y_i = f(x_i) + \varepsilon_i$, becomes $z_i$, in iteratively reweighted least squares (IRLS). So, at each IRLS iteration we fit the additive model to the pseudo-response $z$ against $x$ and the working weights.

The smooth is a technique which allows a data-driven approach to the model-driven model process [29]. The objective function of the model could be written as a minimizer of the objective function as:

$$S(f) = \sum_{i=1}^{n} (y_i - f(x_i))^2 + \lambda \int_a^b \{f''(x)\}^2 dx, \tag{2}$$

where $f(x_i)$ is a smooth function, $\lambda$ is a smoothing parameter, which was used to control the model's smoothness, $f''(x)$ is the second derivative of smooth function to penalize for wiggiliness of $f(x_i)$.

B-spline basis function is used as a basis for the vector space. So cubic spline will be written as a linear combination of those basis function. The objective function of the cubic smoothing spline, based on the penalized least square, and based on Eq 2 is written as:

$$s(f) = (y - f)^T \Sigma^{-1} (y - f) + \lambda f^T k f \tag{3}$$

where k is a roughness penalty matrix, and after setting its derivative to 0 we have:

$$\hat{f} = S(\lambda)y \qquad 4$$

where the influence matrix of $S(\lambda)$ is:

$$S(\lambda) = (I_n + \lambda \Sigma K)^{-1} \qquad 5$$

where $\Sigma$ is written based on the weights as $\Sigma = W^{-1}$. From Eq 5, it is clear that the vector splines do not depend on $y$, and $\lambda$ is fixed so it is a linear smoother.

The process could be summarized as a projection into the linear fit space, where we consider and arrange all parameters based on the smooth vector as [31]

$$f_{(j)k}^*(x_k) = \beta_{(j)k}^* x_k + g_{(j)k}^*(x_k) \qquad 6$$

where $k$ is an index for an explanatory variable and $j$ the index for a random parameter. Superscript is because we just considered variables related to smooth parameters. The residual is formed so the vector back fitting could be conducted for estimation of $g_{(j)k}(x_k)$ as:

$$r_i = z_i - \sum_{k=1}^{p} \hat{\beta}_k x_{ik} \qquad 7$$

The process is summarized as going through the projection, while considering all parameters based on the current smooth parameter, and then non-projection process by considering the original values, while fitting a univariate smooth function.

The above discussion could be linked to the RHS of Eq 2 where we have both the original dataset and the projected values, look at the RHS of Eq 2. First, we estimate the linear parts and the non-linear part of the component function, RHS of Eq 2, or Eqs 6 and 7. By knowing the linear part of the additive model, the partial residual could be solved for estimation of the smooth function.

Correlation is obtained from variance-covariance matrix, which itself is based on inverting the Cholesky of the weight, where weight is part of the IRLS process. Finally, it is important to note that models are considered to account for associations, and not causations across various considered models.

## Results

The results section is presented into 3 subsections based on the assigned categories presented in Table 2, and related variables.

### Students' characteristics

The next few paragraphs discuss associations between various students' characteristics and negative emotional feelings.

**Gender.** The results highlighted that female student experienced higher levels of being afraid, irritable, stressed, and guilty. Gender-specific differences across emotions are expected. For instance, the differences in gender, in terms of stress and fear are investigated in the past studies [32]. Also, another study highlighted while men report more powerful emotions (e.g., anger), women report more powerless emotions such as sadness or fear [33]. In addition, a longer irritability was reported for women compared with men [18].Higher education

The results, see Table 2, revealed that those students doing higher educational studies feel guiltier regarding COVID-19, while feeling less irritable. At the same time, they are more

**Table 2. Results of the SUR for associations between emotions and other factors in a unified model.**

| Coef. | Sad | SE | p-value | Afraid | SE | p-value | Irritable | SE | p-value | Stressed | SE | p-value | Guilty | SE | p-value |
|---|---|---|---|---|---|---|---|---|---|---|---|---|---|---|---|
| (Intercept) | 19.68 | 5.32 | <0.005 | -2.15 | -1.11 | 0.05 | 50.81 | 0.84 | <0.005 | 31.13 | 0.857 | <0.005 | 10.00 | 0.46 | 0.00 |
| | | | | | | | *Students' characteristics* | | | | | | | | |
| Gender | 6.93 | 4.995 | 0.17 | **6.61** | **1.34** | **<0.005** | **2.39** | **0.661** | **<0.005** | **6.45** | **0.997** | **<0.005** | **6.63** | **1.114** | **<0.005** |
| Respondents' highest level of education | 0.67 | 6.058 | 0.91 | **3.36** | **1.06** | **<0.005** | **-2.84** | **0.769** | **<0.005** | -0.16 | 0.665 | 0.81 | **2.57** | **1.047** | **0.01** |
| Age | 5.27 | 4.784 | 0.27 | **3.38** | **1.23** | **0.01** | **10.50** | **0.422** | **<0.005** | **4.80** | **0.625** | **<0.005** | 1.42 | 1.269 | 0.26 |
| Health in general | -1.38 | 5.565 | 0.80 | **-2.07** | **0.49** | **<0.005** | **-2.47** | **0.396** | **<0.005** | **-4.54** | **0.758** | **<0.005** | -1.67 | 1.002 | 0.09 |
| BMI group | -1.19 | 0.989 | 0.23 | -0.43 | 0.46 | 0.36 | **0.98** | **0.48** | **0.04** | 0.36 | 0.598 | 0.55 | -0.92 | 1.166 | 0.43 |
| Family income, relative to others in the US | -1.66 | 1.367 | 0.23 | **-1.46** | **0.62** | **0.02** | **-1.28** | **0.449** | **<0.005** | **-1.68** | **0.437** | **<0.005** | -0.77 | 0.482 | 0.11 |
| | | | | | | | *Precautionary actions* | | | | | | | | |
| Avoid travel | **2.77** | **0.929** | **<0.005** | **2.29** | **0.56** | **<0.005** | **-0.79** | **0.379** | **0.04** | **2.38** | **0.696** | **<0.005** | 0.02 | 0.179 | 0.92 |
| Limit outdoor activities | 1.08 | 1.127 | 0.34 | **1.49** | **0.44** | **<0.005** | 0.22 | 0.441 | 0.62 | **0.68** | **0.276** | **0.01** | 0.25 | 0.168 | 0.14 |
| Precautionary actions: limit exercising at home | -0.06 | 0.89 | 0.95 | 0.68 | 0.52 | 0.19 | 0.15 | 0.429 | 0.73 | 0.28 | 0.259 | 0.28 | **1.32** | **0.204** | **<0.005** |
| Avoiding large group of people | -0.30 | 1.035 | 0.77 | 0.03 | 0.69 | 0.97 | **-2.51** | **0.403** | **<0.005** | -0.50 | 0.314 | 0.11 | **-0.36** | **0.161** | **0.02** |
| Precautionary actions: avoid people outside of your immediate family | 2.20 | 1.2 | 0.07 | **2.79** | **0.64** | **<0.005** | 0.55 | 0.489 | 0.26 | **0.57** | **0.248** | **0.02** | **0.76** | **0.188** | **<0.005** |
| | | | | | | | *Others* | | | | | | | | |
| spend a lot of time thinking about COVID | **6.23** | **1.127** | **<0.005** | **7.60** | **0.78** | **<0.005** | **4.19** | **0.386** | **<0.005** | **7.47** | **0.289** | **<0.005** | **3.53** | **0.513** | **<0.005** |
| Any infected individual among friends/family | **2.65** | **1.079** | **0.01** | **1.54** | **0.72** | **0.03** | **3.81** | **0.953** | **<0.005** | **1.71** | **0.41** | **<0.005** | -0.78 | 0.585 | 0.18 |
| | | | | | | | *smooth* | | | | | | | | |
| Smooth | | DF | P(chi) | | DF | P(chi) | | DF | P(chi) | | DF | P(chi) | | DF | P(chi) |
| S (hours in front of screen) | **-0.31** | **2.9** | **0.02** | -0.35 | 2.9 | 0.46 | 0.10 | 2.9 | 0.13 | -0.11 | 2.9 | 0.51 | **-0.50** | **3** | **0<0.005** |
| S (limiting exercise at the gym) | -0.62 | 2.9 | 0.59 | -0.18 | 2.9 | 0.12 | **0.40** | **3** | **<0.005** | **-0.10** | **2.9** | **<0.005** | 0.52 | 3 | **<0.005** |

* Bolds are significant values at 0.05 significance level.

afraid of COVID-19 compared with other lower educated individuals. It should be reiterated that higher education was considered only from undergraduate to graduate studies.

**Younger age.** Here the binary predictor of age was considered and analyzed. The category of 18–25 accounts for the majority of students. The results highlighted that younger student experience higher feelings of being afraid, irritable, and stressed. Here the irritability has the highest impact, which is in line with the previous reported findings [34].

**Impact of general health.** A question was asked regarding how students rate their general health. The results highlighted that there are negative associations between general health and being afraid, feelings of sadness, being irritable and stressed due to COVID-19. The results highlight the association between better health with having less negative emotions. Our finding is in line with the previous study which found feelings of unsafety and fear are important factors adversely impacting health [35].

**BMI.** It was found that higher degree of irritability is associated with higher BMI among students. The link between BMI and irritability was evaluated in previous study [36]. The results of that study highlighted a positive correlation between higher BMI and higher levels of irritability. Despite the significant association of BMI and feeling irritable, it could be seen that the associations between BMI and other negative feelings are not significant.

**Family income.** The results highlighted that a higher family income reduce feeling afraid, irritable, and stressed. The impact of family income might be due to a justifiable sense of security, which helps students to handle the negative feelings of pandemic in a more reasonable way.

## Precautionary actions

The next section discusses various precautionary actions including rational or irrational. Various actions that are not related to the pandemic were incorporated to highlight the impact of various negative feelings on irrational choice-making behaviors of students. For instance, while avoiding travel might help to curb the pandemic, limiting exercise at home might be considered an extreme behavior.

**Avoid travel.** The public health authorities limited activities such as recreational traveling due to an increase in the spread of the disease. However, those activities tend to work as means of stress relief. The results found that avoiding recreational travel is positively associated with higher levels of sadness, being afraid, being stressed, while it is negatively associated with feelings of irritability. The results highlight the variation across various negative emotions in response to avoiding travel.

**Limit outdoor activities.** The results highlighted positive associations between feelings of being afraid and stressed and limiting outdoor activities. The positive impact of spending time outdoors such as outdoor walking, outdoor exercise or gardening on the amount of stress has been investigated in the past study [37]. Although the actions of traveling seem to be strongly against social distancing, many outdoor activities are less involved with interacting with other people. The variable was incorporated to study a possible extreme precautionary behavior of students.

**Limit exercise at home.** Although working out at home seems to pose no danger of being infected, it was found that still students with higher amounts of guilt, are likelier to skip working out at home.

**Avoid large groups of people and avoiding all people outside the family.** The results found that belief in avoiding a large group of people, besides its importance in tackling the spread of the virus, is negatively associated with feelings of guilt and feeling irritable. The negative association of guilt might be related to the sense of responsibility that students feel while taking this important precaution. On the other hand, avoiding all people outside the immediate family is positively associated with being sad, afraid, stressed and guilty.

In summary, the results highlighted avoiding large group of people, being associated with being infected by the disease, reduces the impacts of various negative emotions. However, at the same time results suggest the important of having connection with outside the family such as close friends, which are important factors in mitigation of various students negatively.

**Thinking about COVID.** The previous study highlighted that thinking too much about the pandemic could be unhealthy and debilitating [38]. Also, while thinking is expected to keep the individuals safe from possible harm, they could be a sign of clinical anxiety [39]. Our result is in line with the literature highlighted that there are positive associations between thinking a lot about COVID and all included feelings.

**Having an infected member in the family.** Having an infected individual in the family increases negative feelings, except for feeling guilty. Here feeling sad might be related to feeling empathy for a friend or family member. Also, feeling afraid might be linked to the possibility of the illness spreading to other family members or losing those infected. On the other hand, feeling irritable might be due to the sense of duty to do something about the problem by assisting society in some way.

Finally, it should be noted that we smoothed two variables of time spent in front of a screen and whether students limited their exercise at the gym or not. That was because the impacts of those variables primarily were found to be non-linear, *p*-value>0.1, level of significance, while after smoothing them, they were found to be important.

To summarize the findings, Fig 1A, 1B and 1C are provided to give the signs of various associations. As can be seen from those figures, there are significant variations across different emotions. Fig 1A presents lack of significant across the majority of predictors and feeling sad which is different from feeling afraid. For instance, Fig 1A shows while there is a negative association between being afraid and family income, the association between that predictor and feeling sad is not significant. The similar explanation applies for Fig 1B. For instance, while there is no significant association between being afraid, being sad and limited exercise at home, the association of that predictor and feeling guilty is positive.

Regarding Fig 1C, there are some variations between feeling irritable and stressed. For instance, while there is a positive association between feeling stressed and avoiding outside immediate family, that association is not significant for feeling irritable. To highlight the positive, negative or no associations in Fig 1, "+", "-"and "×"are incorporated on the arrows to highlight associations with different signs.

## Correlation across models' error terms

Correlation was obtained from variance covariance matrices, where that metric itself, was obtained by inverting the working weights, see Table 3. Here the correlation across error terms of three models were found to be considerable, which is addressed in the next paragraph

The correlation across the model's error terms of stress and afraid was found to be 0.47. It is in line with a previous study, which investigated the relationship between fear and stress on

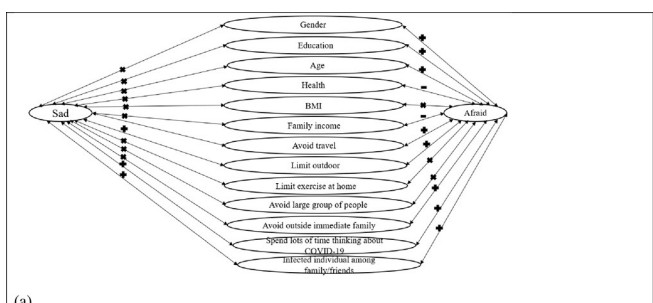

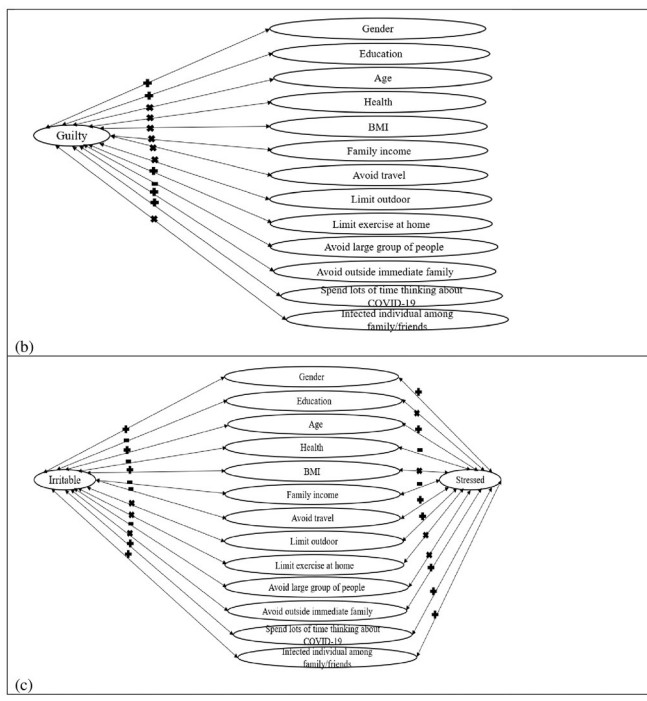

**Fig 1. Associations between various negative emotions and students behaviors, and characteristics, positive, negative and no associations are highlighted by "+","-", and "×".**

**Table 3. Correlations across models' error terms.**

|  | Afraid | Irritable | Stress | Guilty |
|---|---|---|---|---|
| Sad | 0.41 | 0.301 | 0.465 | 0.26469 |
| Afraid | ----------- | 0.143 | 0.47 | 0.2457 |
| Irritable | --------- | --------- | 0.36 | 0.12 |
| Stress | --------- | --------- | ------ | 0.213 |

the immigrant family [40]. The correlation between feeling sad and afraid due to COVID-19 was found to be 0.41. That is in line with the previous study, which found an association between feelings of fear and sadness in the lives of adolescents [41]. The fear in that study was related to the fear of not surviving the negative feelings, or fear of not getting help.

## Discussion

Emotions are deep-rooted in every aspect of human beings' lives. The choices we make are aided by emotions, where emotions, themselves, are elicited during considering various alternatives as being disadvantages or advantages (cost-benefit analysis) [42]. Also, prevalence of negative emotions may turn into chronic stress or other disorders such as depression or anxiety.

In the case of college students impacted by a pandemic, studying various emotions are especially important as emotions are associated with mental health and various decision-making behaviors. A deeper understanding regarding behavioral characteristics being associated with various negative emotions during a life-changing COVID-19 pandemic could be a useful insight.

Our associated results of lower irritability and higher education are in line with the previous study, which showed that higher educational level can reduce negative emotions, compared with their lower educated counterpart [43]. In general, it seems that more educated people take the pandemic more seriously than lower educated individuals, which is highlighted by feeling more afraid.

They also feel more responsible which might be reflected in higher feeling of guilt.

We found that younger students experience higher feelings of being afraid, irritable, and stressed. The impact might be due to the experience and control that the older group has over the feelings associated with the pandemic: the older age has a wider horizon, and an increased awareness regarding life challenges and how to overcome them.

It is expected that various irrational behaviors associated with COVI-19 are caused by lack of understanding about the pandemic and the impact of related negative emotions. "Catastrophizing" has been defined as a tendency to magnify the imposed threats [44]. That is an overreaction of people to unprecedented stressor or life events. Also, that lead to dysfunctional negative behaviors or feelings [44]. It has been highlighted that individuals under various negative emotions, e.g., anxiety, systematically distort the meaning of events so they interpreted their experiences about an event in a negative and self-defeating way [7].

Our findings of behaviors such as limiting exercise at home, or even limiting outdoor activities are expected to be associated with those negative emotions, which result in magnifying or distorting the meaning of the threat.

Especially we found that there is a positive association between limiting exercise at home and feeling guilty. Feeling of guilt could be defined as a function of monitoring one's own behaviors in relation to others [45]. Also, that has been highlighted as a sense of responsibility for others and the acknowledgement of the action and call for amendment [46]. So, it is

expected that limiting exercise at home, might work as a means of self-punishment or posing a self-suffering to share with other victims, suffering from the pandemic [47]. Timely clarification of a possible misconception by the media, and surveillance of student's emotions and behaviors are recommended at the time of a pandemic, or any unprecedented events so appropriate actions could be made.

One of the interesting aspects of the human being is the ability to adapt under challenging and difficult circumstances [48]. However, sometimes under misconception, or due to various emotions resulting from other aspects of a pandemic, individuals take behaviors that negatively impact their mental health. More policies and educational programs are needed to help students to adopt the right approach by increasing their awareness, by helping them to focus on active problem solving, and practicing mindfulness.

Avoiding a large group of people in curbing the pandemic might sound similar to "avoiding all people outside the immediate family". However, our results highlighted that avoiding all people outside the immediate family is directly associated with negative emotions.

The take-away could be linked to the previous study [49], where a contrast was made between physical distancing and social distancing during COVID-19. The study discussed the implication of the term "social distancing" which might have detrimental effects. That is because it might invoke negative feelings of being ignored, unwelcomed, or left alone with fear. More educational programs are needed to encourage students to feel a sense of responsibility by staying away from a large population but keeping in touch with close friends while keeping safety in mind.

The negative association between anger/irritability and travel might be provoked by attack or danger [50], so by avoiding traveling, students might feel less exposed to the threat and consequently feel less irritable.

As mentioned before, feeling irritable is mainly related to lack of certainty about the future. By taking necessary precautions, students will be protected from getting infected as well as gaining a stronger sense of certainty about the future. Having an infected person in the community along with past experiences with similar situations, might steer individuals to different decision-making processes [51].

Based on the past experience, it is reasonable to expect that having infected family member might change the belief of individuals regarding the severity and magnitude of the decease. The public response to the pandemic is not always aligned with science, so more educational programs and research are needed to educate students on the consequences of being infected.

Our results highlighted that higher income is associated with various negative emotions such as being afraid, irritable, and stress.

It has been studied that instability through dramatic disruption, or financial challenges can impact individuals' wellbeing and the level of stress they sustain [52]. Universities play a pivotal role in helping students to manage their finances. That could be done by providing an appropriate source of income through part-time careers. This is particularly important since many of students lost their jobs and source of income during this pandemic.

Finally, our results highlighted that while there are mainly similarities across various emotions in terms of signs, they are different in terms of magnitudes and signs for other predictors. For instance, while higher education is negatively associated with higher feeling of irritability, it is positively associated with feelings of being guilty and afraid. Or while, avoiding travel is positively associated with being stressed, sad and afraid, it is negatively associated with being irritable. The findings were not achievable in case of investigating the emotion individually, and implementation of a unified model was needed to obtain those findings.

Due to the universities closure, we used a convenience sampling (or nonprobability sampling) method, which may affect the results. It means the profile of this sample may be a biased

representation of the US college students in general. Therefore, a precaution should be given while interpreting and using the findings. In addition, only specific locations with unique demographic characteristics, e.g., location, weather, climate, were incorporated. Emotions and behaviors are expected to be complex [53], thus other possible factors related to the above points should be considered. As the results of the survey were collected in spring 2020 when the first and most severe real lockdown occurred in the U.S., it is not practical to recollect the data to incorporate more observations.

It is worth noting that, having students from other parts of the country—such as southern states of the US—could have brought more reliable outcome for present study. Therefore, in order to make sure about the generalizability of these results, future research could focus on double checking the results with larger and randomized sample of students from more states and universities.

For future direction of research, more investigations such as in-person interviews can be considered to highlight the justification behind taking unnecessary precautionary behaviors, e.g., limited exercise at home. That way, our hypothesis regarding impaired emotions of students could have been confirmed. Despite the limitations, we were able to highlight the associations between various variables and negative emotions, especially during life-changing pandemic of COVID-19.

## Conclusion

In general, the present study brings us some important information about the various students' emotions and their relationship with other factors. As such, this study gives a panoramic vision about the mental well-being of students and their causes, which can help educators, university and health authorities to assist the vulnerable and affected groups of students during the post-pandemic. The results highlighted that being female and older, having a higher education, having poor health and higher BMI exacerbate various negative emotions at the time of the pandemic. It means, these groups of students can be considered as much affected group of students by negative emotions during the pandemic.

Besides, spending too much thinking about COVID-19 was also found to increase various negative emotions, while higher family income result in a reduction of feeling stressed and irritable. On the other hand, having an infected individual in the community was found to increase negative feelings possibly due to enhancing perceptions regarding the consequences of having COVID-19.

In terms of precautionary behaviors, we incorporated two types of behaviors, rational and irrational. For instance, limiting outdoor activities was positively associated with various negative emotions. Similarly, limiting socializing with close friends was found to be associated with various negative emotions. However, behaviors such as avoiding a large group of people was found to reduce the associated negative feelings.

Based on the above discussion, students should hear the truth about the gravity of the disease and while taking scientific precautions seriously they should not let negative emotions control their behaviors and take it to the extreme so as to negatively impact their mental health. The painful insights from COVID-19 could be employed not only for the current pandemic but for future outbreaks.

## Author Contributions

**Data curation:** Mahdi Rezapour.

**Investigation:** Arash Dehzangi, Farzaneh Saadati.

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
