## [Decision Letter · Decision Letter 0]

19 Dec 2021

PONE-D-21-32396Emotions and students rational and irrational behaviors, a Covid-19 case studyPLOS ONE

Dear Dr. Rezapour,

Thank you for submitting your manuscript to PLOS ONE. After careful consideration, we feel that it has merit but does not fully meet PLOS ONE’s publication criteria as it currently stands. Therefore, we invite you to submit a revised version of the manuscript that addresses the points raised during the review process.

We look forward to receiving your revised manuscript.

Kind regards,

Sanjay Kumar Singh Patel, Ph.D.

Academic Editor

PLOS ONE

Journal Requirements:

(There is no funding!)

Reviewers' comments:

Reviewer's Responses to Questions

**Comments to the Author**

1. Is the manuscript technically sound, and do the data support the conclusions?

Reviewer #1: Yes

Reviewer #2: Yes

Reviewer #3: Partly

2. Has the statistical analysis been performed appropriately and rigorously? 

Reviewer #1: Yes

Reviewer #2: Yes

Reviewer #3: Yes

3. Have the authors made all data underlying the findings in their manuscript fully available?

Reviewer #1: Yes

Reviewer #2: Yes

Reviewer #3: Yes

4. Is the manuscript presented in an intelligible fashion and written in standard English?

Reviewer #1: Yes

Reviewer #2: Yes

Reviewer #3: Yes

5. Review Comments to the Author

Reviewer #1: In the current research article entitled "Emotions and students rational and irrational behaviors, a Covid-19 case study", by Rezapour, conducted by using the results of a survey conducted across 2,534 students enrolled in 7 states in the US. Various feelings such as sad, irritable, stress and guilt were analyzed in a unified model by means of seemingly unrelated regression. it was found that having a higher body mass index (BMI), self-rated worse health conditions, and limiting outdoor activities would have detrimental effects on the mental health of students. This article addresses a research topic of great interest, which is under intense investigation in the past years and the manuscript is generally well-written. However, this reviewer has certain suggestions that would help produce a more comprehensive overview of the topic:

Suggestions: -

1. The authors should cross-check all abbreviations in the manuscript. Initially, define in full name followed by abbreviation.

2. The English of manuscript can be polished (minor).

3. The authors may additionally provide one Figure as summary, challenges, or prospect of the present study.

Reviewer #2: In this paper entitled " Emotions and students rational and irrational behaviors, a Covid-19 case study", the author's goal is to conduct an in-depth evaluation of associations between various emotions that students experience during COVID-19 pandemic". The manuscript is easy to understand and data-oriented. However, it requires few corrections.

Minor Comments:

1) The English may be polished. There are grammatical errors and spelling mistakes in the manuscript.

2) Discuss studies related to the manuscript in the introduction.

3) What are the criteria for selecting the emotions incorporated in the study.

4) Discussion: The information about mortality rate and various prevention approaches should be provided related to immunity, and health, and influence of COVID-19 variants i.e. doi: 10.1007/s12088-020-00908-0; doi: 10.1007/s12088-020-00893-4, and doi: 10.1007/s15010-021-01734-2.

5) Mentions the tools used during the dataset analysis in the Data section. Also, highlight how the present manuscript is different from studies done using the same data set.

6) The author should provide at least one or two illustrations (additional Figures) to highlight the summary and significance.

7) Advanced statistical tools can be used to analyze the study comprehensively.

Reviewer #3: Summary:

The authors report a case study titled “Emotions and students rational and irrational behaviors, a Covid-19 case study”. This study uses a survey-based approach along with statistical analyses to analyze the effects of Covid-19 pandemic on the mental health of students. A total of 2534 students participated in this study. Feelings such as sadness, irritability, stress and guilt were analyzed in this study. The topic of study is very relevant and interesting to the general audience.

Major Comment:

According to the National Center for Education Statistics (NCES), 19.7 million students were enrolled at U.S. colleges in fall 2020. Among them there were approximately 16.7 million undergraduate students and 3.1 million graduate students. States like California, Texas, Illinois, Minnesota enrolls a significant population of students. Additionally, these states represent the diverse demographics (location, weather, climate etc.) of the USA, which could influence the emotions among students. Unfortunately, the authors have not provided any data from these states. Assessing the emotions and behavior could be complex and hence several factors need to be considered. A study by Kim et al. (10.3390/ijerph14040431) suggested that both demographic and environmental factors can influence mental health which translates to behavior. In light of the above reasons, I believe the current study is limited by the number of students participated as well as location. The authors could include more participants from schools of Mid-western states and bay areas (like California).

6. PLOS authors have the option to publish the peer review history of their article (what does this mean?). If published, this will include your full peer review and any attached files.

Reviewer #1: **Yes: **Vinay Kumar

Reviewer #2: **Yes: **Aditya Kumar Sharma

Reviewer #3: **Yes: **Dibyendu Mondal

---

## [Decision Letter · Decision Letter 1]

22 Feb 2022

Students negative emotions and their rational and irrational behaviors during COVID-19 outbreak

PONE-D-21-32396R1

Dear Dr. Rezapour,

We’re pleased to inform you that your manuscript has been judged scientifically suitable for publication and will be formally accepted for publication once it meets all outstanding technical requirements.

Kind regards,

Academic Editor

PLOS ONE

Additional Editor Comments (optional):

Reviewers' comments:

Reviewer's Responses to Questions

**Comments to the Author**

1. If the authors have adequately addressed your comments raised in a previous round of review and you feel that this manuscript is now acceptable for publication, you may indicate that here to bypass the “Comments to the Author” section, enter your conflict of interest statement in the “Confidential to Editor” section, and submit your "Accept" recommendation.

Reviewer #1: All comments have been addressed

Reviewer #3: All comments have been addressed

2. Is the manuscript technically sound, and do the data support the conclusions?

Reviewer #1: Yes

Reviewer #3: Yes

3. Has the statistical analysis been performed appropriately and rigorously? 

Reviewer #1: Yes

Reviewer #3: Yes

4. Have the authors made all data underlying the findings in their manuscript fully available?

Reviewer #1: Yes

Reviewer #3: Yes

5. Is the manuscript presented in an intelligible fashion and written in standard English?

Reviewer #1: Yes

Reviewer #3: Yes

6. Review Comments to the Author

Reviewer #1: Authors have incorporated all comments to the manuscript "Students negative emotions and their rational and irrational behaviors during COVID-19 outbreak"

Reviewer #3: (No Response)

7. PLOS authors have the option to publish the peer review history of their article (what does this mean?). If published, this will include your full peer review and any attached files.

Reviewer #1: **Yes: **Vinay Kumar

Reviewer #3: No

---

## [Editor Report · Acceptance letter]

24 Feb 2022

PONE-D-21-32396R1 

Students’ negative emotions and their rational and irrational behaviors during COVID-19 outbreak 

Dear Dr. Rezapour:

I'm pleased to inform you that your manuscript has been deemed suitable for publication in PLOS ONE. Congratulations! Your manuscript is now with our production department. 

Kind regards, 

on behalf of

Dr. Robert Jeenchen Chen 

Academic Editor

PLOS ONE